# Increased Reflux Esophagitis after *Helicobacter pylori* Eradication Therapy in Cases Undergoing Endoscopic Submucosal Dissection for Early Gastric Cancer

**DOI:** 10.3390/cancers13081779

**Published:** 2021-04-08

**Authors:** Masaki Katsurahara, Ichiro Imoto, Yuhei Umeda, Hiroshi Miura, Junya Tsuboi, Reiko Yamada, Taro Yasuma, Misaki Nakamura, Yasuhiko Hamada, Hiroyuki Inoue, Kyosuke Tanaka, Noriyuki Horiki, Esteban C. Gabazza, Yoshiyuki Takei

**Affiliations:** 1Department of Endoscopic Medicine, Mie University and Graduate School of Medicine, Edobashi 2-174, Tsu, Mie 514-8507, Japan; mkatura@clin.medic.mie-u.ac.jp (M.K.); kyosuket@clin.medic.mie-u.ac.jp (K.T.); nohoriki@clin.medic.mie-u.ac.jp (N.H.); 2Digestive Endoscopy Center, Doshinkai Tohyama Hospital, Minami-Shinmachi 17-22, Tsu, Mie 514-0043, Japan; hszq5wxb@ztv.ne.jp; 3Department of Gastroenterology and Hepatology, Mie University Faculty and Graduate School of Medicine, Edobashi 2-174, Tsu, Mie 514-8507, Japan; umedarf1128@clin.medic.mie-u.ac.jp (Y.U.); h-miura@clin.medic.mie-u.ac.jp (H.M.); t-junya@clin.medic.mie-u.ac.jp (J.T.); reiko-t@clin.medic.mie-u.ac.jp (R.Y.); misa4944@clin.medic.mie-u.ac.jp (M.N.); y-hamada@clin.medic.mie-u.ac.jp (Y.H.); hiro1024@clin.medic.mie-u.ac.jp (H.I.); ytakei@clin.medic.mie-u.ac.jp (Y.T.); 4Department of Immunology, Mie University Faculty and Graduate School of Medicine, Edobashi 2-174, Tsu, Mie 514-8507, Japan; t-yasuma0630@clin.medic.mie-u.ac.jp

**Keywords:** *Helicobacter pylori*, reflux esophagitis, early gastric cancer, gastritis, Japanese population

## Abstract

**Simple Summary:**

*Helicobacter pylori* infection is associated with the development of gastric cancer. Reflux esophagitis may occur in the postoperative period of patients undergoing surgical therapy for gastric cancer. The role of eradication therapy of *Helicobacter pylori* in reflux esophagitis is controversial. Here, we evaluated the occurrence of reflux esophagitis before and after *Helicobacter pylori* eradication in patients having endoscopic submucosal resection for early gastric cancer. Reflux esophagitis before and after eradication therapy was evaluated during the follow-up. While reflux esophagitis incidence increased from 3.1% to 18.8% in the successful eradication group, no case of reflux esophagitis was observed in the failed eradication group. There was a significant correlation between successful *Helicobacter pylori* eradication rate and reflux esophagitis development. This study demonstrates that a successful *Helicobacter pylori* eradication therapy is a risk factor for newly developed reflux esophagitis in patients having endoscopic submucosal dissection for early gastric cancer.

**Abstract:**

Background: The role of *Helicobacter pylori* in the pathogenesis of reflux esophagitis is controversial. This study investigated the frequency of reflux esophagitis before and after *H. pylori* eradication in patients having endoscopic submucosal dissection for early gastric cancer. Methods: This study included 160 patients that fulfilled the study’s criteria. Endoscopy was performed before and after *H. pylori* eradication, and reflux esophagitis was evaluated during the follow-up period. Results: Seropositivity for *H. pylori* in patients with early gastric cancer was 68.8%, 101 of them received eradication therapy. During the follow-up period, the incidence of reflux esophagitis increased from 3.1% to 18.8% in the successful eradication group but no case of reflux esophagitis was observed in the failed eradication group. The univariate and multivariate analyses showed a significant correlation between successful *H. pylori* eradication rate and the development of reflux esophagitis. Conclusions: This study demonstrated that a successful *H. pylori* eradication therapy is a risk factor for newly developed reflux esophagitis in patients with endoscopic submucosal dissection for early gastric cancer.

## 1. Introduction

*Helicobacter pylori* (*H. pylori*) infection is the most frequent cause of gastroduodenal ulcerative disease, gastric cancer, and gastric mucosa-associated lymphoid tissue lymphoma [1,2]. Gastroesophageal reflux disease (GERD) is frequently caused by the abnormal reflux of gastric content into the esophagus. GERD diagnosis includes endoscopically proven reflux esophagitis and non-erosive reflux disease, which is characterized by the absence of esophageal mucosal damage during endoscopy, despite the presence of typical symptoms [3]. The development of reflux esophagitis after successful *H. pylori* eradication was first reported by Schutze et al. in 1995 [4]. After that, Labenz et al. reported in a prospective study that the cure of *H. pylori* infection in patients with duodenal ulcer leads to reflux esophagitis [5]. Although subsequent investigators reported contradicting results, the Maastricht III consensus report from the European countries that “*H. pylori* eradication therapy needs not be withheld for fear of provoking reflux esophagitis” underscores the clinical and general importance of this post eradication therapy complication [6,7,8]. A high incidence of reflux esophagitis after successfully eradicating *H. pylori* has been particularly observed in Eastern countries, including Japan [9,10,11]. We have previously shown that post-eradication reflux esophagitis in Japanese patients is significantly associated with the severity of hiatal hernia and a low gastric juice pH [10]. The relatively high incidence of reflux esophagitis after *H. pylori* eradication in the Japanese population has been attributed to the frequent observation of severe gastric mucosal atrophy and reduced gastric acid secretion before *H. pylori* eradication. Hypochlorhydria and gastric mucosal atrophy are also frequently observed in patients with gastric cancer [12]. However, there is no clear information about the incidence of reflux esophagitis after eradicating *H. pylori* in gastric cancer patients. Na et al. reported no increase in the incidence of reflux esophagitis symptoms after *H. pylori* eradication therapy in patients that underwent endoscopic mucosal resection or endoscopic submucosal dissection for gastric neoplasms [13]. However, no endoscopic study was performed to confirm the presence or absence of reflux esophagitis after eradication therapy, and there is no study performed in a homogenous group of patients with early gastric cancer after endoscopic submucosal dissection. In addition, no study has reported potential risk factors for reflux esophagitis after eradication therapy.

The present investigation evaluated the frequency of endoscopically confirmed reflux esophagitis before and after *H. pylori* eradication therapy in patients that underwent endoscopic submucosal dissection for early gastric cancer and the potential risk factors for reflux esophagitis after eradication therapy.

## 2. Materials and Methods

### 2.1. Patients

This study comprised 429 patients with gastric cancer admitted to the Department of Gastroenterology and Hepatology, Mie University Hospital, from January 2006 through December 2016. We included 160 patients (males 122, females 38, mean age 69.7 years, range 37–89 years) that fulfilled the study’s entry criteria. We retrieved the data of the patients from medical records.

### 2.2. Study Design

This clinical investigation was a retrospective single-center study. Endoscopy was performed using a magnifying narrow-band-imaging (NBI) endoscopy (Q240Z, H260Z; Olympus Medical Systems Co. Tokyo, Japan). We obtained informed consent from all patients, and the study was conducted following the Principles of the Helsinki Declaration. The exclusion criteria of the study were as follows: current medication with proton pump inhibitors or H_2_ receptor antagonists during the follow-up study (*n* = 122), lack of follow-up endoscopy (*n* = 74), gastric surgery after endoscopic submucosal dissection (*n* = 46), previous gastric surgery (*n* = 11), or eradication therapy (*n* = 24) (Figure 1). Endoscopic submucosal dissection in early gastric cancer and follow-up by esophagogastroduodenoscopy were the inclusion criteria of the study. The proton-pump inhibitor was administered for three days before and eight weeks after the endoscopic submucosal dissection. We assessed the severity of reflux esophagitis and gastric mucosal atrophy and the presence of hiatal hernia during the first esophagogastroduodenoscopy performed one month before endoscopic submucosal dissection. Esophagogastroduodenoscopy was performed two to three times during the follow-up period. *H. pylori* eradication therapy was started two months after the endoscopic submucosal dissection. The presence and severity of reflux esophagitis were evaluated by esophagogastroduodenoscopy approximately 6 months after eradication therapy in both successful (5.7 ± 2.4 months) and failed (5.6 ± 2.4 months) eradication groups. Another esophagogastroduodenoscopy was performed one year after eradicating *H. pylori* infection.

After completing the *H. pylori* eradication therapy, we compared the endoscopic findings of the first, second, and third esophagogastroduodenoscopy to determine reflux esophagitis’ clinical progression. The severity of reflux esophagitis was grouped according to the Los Angeles classification, and the grade of hiatal hernia followed the Hill’s gastroesophageal flap valve (GEFV; I to IV) classification [14]. The grade of gastric mucosal atrophy was based on the classification of Kimura et al. [15].

For the first-line *H. pylori* eradication therapy, the patients received the standard 7-day triple therapy that included proton-pump inhibitors (lansoprazole 30 mg, omeprazole 20 mg, rabeprazole 10 mg, or esomeprazole 20 mg, twice a day), clarithromycin (200 mg, twice a day), and amoxicillin (750 mg, twice a day). Second-line therapy with a proton-pump inhibitor, amoxicillin 750 mg, and metronidazole 250 mg twice a day was indicated in unresponsive cases. A third-line and a fourth-line therapy were also indicated in persistently unresponsive cases. Patients underwent esophagogastroduodenoscopy to evaluate the presence of reflux esophagitis approximately 5 months after the second-line therapy, 10 months after the third-line therapy, and 6 months after the fourth-line eradication therapy.

The *H. pylori* status was evaluated by measuring the serum anti-IgG antibody levels using a commercial enzyme immunoassay kit (E-plate Eiken *H. pylori* antibody II kit, Eiken Chemical Co., Ltd.) before the endoscopy submucosal dissection. The measurable titers were ≥3 U/mL and <100 U/mL. Antibody titer above a cut-off level of ≧10.0 U/mL was regarded as *H. pylori*-positive. The *H. pylori* fecal antigen test or the ^13^C-urea breath test was carried out one month after therapy completion to determine the eradication therapy’s response.

### 2.3. Statistical Analysis

The difference between groups was evaluated by the chi-squared test and Mann–Whitney U-test. Odds ratios were calculated by multiple logistic regression analyses. A *p* < 0.05 was considered statistically significant. The SPSS version 26 statistical software was used for the statistical analysis [3]. 

## 3. Results

### 3.1. Patient Characteristics

*H. pylori* seropositivity (titer ≧ 10.0 U/ml) was 68.8% (110/160) in patients with endoscopic submucosal dissection for early gastric cancer. Out of 110 patients with *H. pylori* infection, 101 patients received *H. pylori* eradication therapy, whereas nine patients refused to receive therapy (Figure 1). After the first-line therapy, the successful eradication rate was 77.2%, and after the second-line therapy, it was 71.4%. The overall successful eradication rate was 95.0% when the third-line and fourth-line therapy were included (Table 1). All patients (*n* = 160) included in the present study were allocated into the following groups: a group with successful eradication (*n* = 96), a failed eradication group (*n* = 5), and an untreated control group (*n* = 59). This control group included seropositive patients that refused treatment (*n* = 9) and untreated *H. pylori*-seronegative patients (*n* = 50) (Table 2, Figure 1). The frequency of gastric mucosal atrophy and the age, sex, body mass index, the number of cases with hiatal hernia, medical history of diabetes mellitus, hypertension, hyperlipidemia, intake of alcohol or nonsteroidal anti-inflammatory drugs (NSAIDs) were not different between successful eradication and failed eradication groups (Table 2).

### 3.2. Frequency and Severity of Reflux Esophagitis before and after Therapy

The frequency of reflux esophagitis in the group of patients with successful eradication increased from 3.1% (3/96)to 18.8% (18/96). Among the 18 patients, reflux esophagitis was grade A in 12 and grade B in 6. Three cases in the group with successful eradication therapy had pre-existing reflux esophagitis. Excluding these three cases, the frequency of newly developed reflux esophagitis was 16.1% (15/93) in the group with successful eradication therapy.

None of the patients with failed eradication therapy (*n* = 5) developed reflux esophagitis. Four (6.7%) patients (4/59) of the untreated control group had (pre-existing) reflux esophagitis of grade A during the first esophagogastroduodenoscopy, and two (3.3%) patients (2/59) developed reflux esophagitis (grade A or grade B) during the follow-up study. 

There was a significant difference (*p* = 0.014) in the incidence of newly developed reflux esophagitis when the successful eradication group was compared with the failed eradication and untreated control groups. The grade of reflux esophagitis after eradication was mild (grade A–B) in all groups.

### 3.3. Univariate and Multivariate Analysis

The successful *H. pylori* eradication rate was significantly correlated with newly developed reflux esophagitis by univariate and multivariate analyses (Table 3).

## 4. Discussion

Reflux esophagitis is defined as a mucosal injury of the lower esophagus induced by exposure to gastric juice. Patients with the disease generally have adequate gastric acid secretion with impairment of the esophageal defense mechanism against reflux contents. Although the infection rate by *H. pylori* has declined in the young population of Japan in recent years, the number of people with increased gastric acid secretion has incremented [16,17]. This condition has led to an increased prevalence of reflux esophagitis since the end of 1990. Furukawa et al. reported a reflux esophagitis prevalence of 16.3% in 6010 patients undergoing endoscopy [18]. Fujiwara et al. reported a reflux esophagitis prevalence of 10.6% in outpatients and a mean prevalence of 7.6% in subjects undergoing health-checkup [19]. Here, we found that only 4.4% of patients having endoscopic submucosal dissection for early gastric cancer have reflux esophagitis. Reflux esophagitis is essentially an acid-related disease, and gastric cancer frequently occurs in an atrophic gastric mucosa with decreased acid secretion [20]. This may explain the low prevalence of reflux esophagitis in our patients compared to previous studies. 

Since Schutze et al., many studies have reported the association of *H. pylori* eradication with reflux esophagitis or GERD [4]. However, subsequent studies have shown contradicting results [21,22,23,24]. Laine et al. reported in eight double-blind prospective studies that the incidence of reflux esophagitis is not increased in duodenal ulcer patients with successful *H. pylori* eradication compared to patients with persistent *H. pylori* infection [21]. In addition, a meta-analysis showed that the eradication of *H. pylori* exerts no significant effect on the development of reflux esophagitis in the short- or long-term and that a successful *H. pylori* eradication is not associated with reflux esophagitis [22,23]. Notwithstanding, a meta-analysis reported by Xie et al. showed a significantly increased risk of GERD in patients with successful eradication therapy compared to patients with non-successful eradication [24]. Further, another recent meta-analysis by Sugimoto et al. suggested that eradication therapy increased the risk of reflux esophagitis diagnosed by endoscopy despite the absence of reflux-related symptoms [25]. These discrepant reports may be explained by several factors, including age, race, *H. pylori* virulence, and presence of gastric atrophy, gastric or duodenal ulcer. For example, contrary to results reported from Japan, duodenal ulcers are more frequent than gastric ulcers in European countries. El-Omar et al. reported that eradicating *H. pylori* infection reduces gastrin-mediated acid secretion by two-thirds in duodenal ulcer patients [26,27]. This low gastrin secretion may explain why reflux esophagitis is less likely to occur after *H. pylori* eradication in cases with duodenal ulcers. Another potential confounding factor is the virulence of the *H. pylori* strain. Most *H. pylori* strains infecting East Asian populations are cagA-positive and vacA s1m1-type. This type of *H. pylori* is highly virulent [28]. Infection with virulent *H. pylori* strains induces much more severe gastric mucosal inflammation with hypochlorhydria and increases corpus gastritis and gastric cancer risk. Post-eradication recovery of acid secretion may explain reflux esophagitis in patients with severe corpus gastritis. 

The reported frequency of post-eradication reflux esophagitis in the Japanese population ranges between 15% and 30%. Hamada et al. reported an estimated prevalence of reflux esophagitis of 18% in gastritis or peptic ulcer patients after eradication therapy compared to 0.3% in patients with no eradication therapy [9]. In a similar study, Inoue et al. reported a prevalence of reflux esophagitis within 12 months of 20.5% in Japanese patients with successful eradication and 3.8% in patients with failed therapy [10]. Take et al. followed 1187 patients with peptic ulcer disease and found that 27.9% of them developed reflux esophagitis after eradicating *H. pylori* infection, and 13.9% had persistent infection [11]. Scarce information is available on the frequency of reflux esophagitis after eradication in patients with underlying gastric cancer. Na et al. reported no GERD symptoms after *H. pylori* eradication therapy in patients undergoing endoscopic resection for gastric cancer or adenoma [13]. However, in the present study, we detected a significant incidence of reflux esophagitis by endoscopic study after successfully eradicating *H. pylori* infection in patients that underwent endoscopic submucosal dissection for early gastric cancer. After evaluating the role of confounding factors by univariate and multivariate analysis, we found that successful eradication therapy is the only factor associated with reflux esophagitis development. 

## 5. Conclusions

In conclusion, this study shows (1) a low frequency of reflux esophagitis in patients that underwent endoscopic submucosal dissection for early gastric cancer before eradication therapy, (2) a higher incidence of reflux esophagitis in patients with successful eradication therapy than in patients with failed therapy, and that (3) eradication therapy is an independent risk factor for reflux esophagitis.

## Figures and Tables

**Figure 1 cancers-13-01779-f001:**
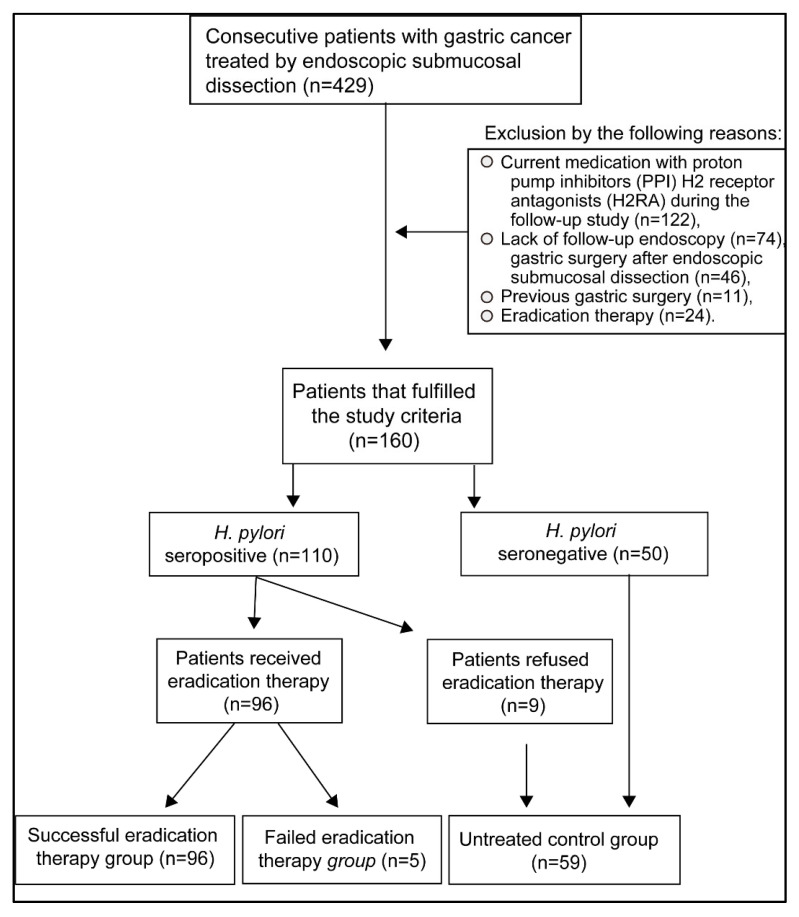
**Study selection criteria.** The records of 429 patients with early gastric cancer were evaluated. Patients that fulfilled the entry criteria were selected.

**Table 1 cancers-13-01779-t001:** The successful eradication rate of *Helicobacter pylori* after different lines of therapy.

	Successful Eradication Group	%
**First-line therapy**	78/101	77.2%
**Second-line therapy**	15/21	71.4%
**Third-line therapy**	2/3	66.7%
**Fourth-line therapy**	1/1	100%
**Total**	96/101	95.0%

**Table 2 cancers-13-01779-t002:** Patients’ characteristics.

Variables	Successful Eradication Group (*n* = 96)	Failed Eradication Group (*n* = 5)	*p* Values *	Untreated Control Group (*n* = 59)	*p* Values *
Age (years)	69.4 ± 9.1	71.4 ± 5.1	0.610	69.9 ± 7.4	0.848
Sex (male/female)	73/23	5/0	0.213	44/15	0.837
Body mass index (kg/m^2^)	22.6 ± 3.1	22.2 ± 2.6	0.803	23.0 ± 3.3	0.381
Hiatus hernia (presence/absence)	56/40	3/2	0.941	38/21	0.452
Gastritis (close/open)	18/78	1/4	0.944	21/38	0.019
Diabetes (yes/no/unknown)	10/83/3	1/4	0.523	12/47	0.102
Hypertension (yes/no/unknown)	37/56/3	2/3	0.992	31/28	0.123
Hyperlipidemia (yes/no/unknown)	16/73/3	0/5	0.311	15/44	0.220
Alcohol (yes/no/unknown)	33/58/5	3/2	0.286	20/38/1	0.662
Smoking (yes/no/unknown)	10/81/5	0/5	0.434	2/57/0	0.094
NSAID use (yes/no/unknown)	5/86/5	0/5	0.590	7/52/0	0.160

* compared to successful eradication group.

**Table 3 cancers-13-01779-t003:** Univariate and multivariate analyses.

Univariate Analysis	Odds Ratio	95% Confidence Interval	*p* Values
Age (<70)	2.781	0.929–8.324	0.067
Sex	1.041	0.318–3.413	0.947
Body mass index (≧25)	0.436	0.094–2.014	0.287
Hiatus hernia	0.741	0.269–2.039	0.561
Gastritis	0.738	0.242–2.254	0.594
Diabetes	0.359	0.0425–2.863	0.333
Hypertension	1.041	0.240–2.026	0.507
Hyperlipidemia	0.436	0.604–5.907	0.275
Alcohol	0.741	0180–1.974	0.397
Smoking	0.738	0.095–6.603	0.83
NSAID use	0.359	0.374–9.583	0.441
Successful eradication	5.577	1.227–25.347	0.026
**Multivariate analyses**			
Successful eradication	5.887	1.238–28.001	0.026

## Data Availability

Data are available upon reasonable request.

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
