# Peer review of "Increased Reflux Esophagitis after Helicobacter pylori Eradication Therapy in Cases Undergoing Endoscopic Submucosal Dissection for Early Gastric Cancer"

_cancers, 2021, doi:10.3390/cancers13081779_

Round 1
Reviewer 1 Report
The following are my comments and recommendations for Drs. Katsurahara and Imoto, on their paper on ‘Increased reflux esophagitis after Helicobacter pylori eradication therapy in cases undergoing endoscopic submucosal dissection for early gastric cancer’.
These results were interesting, but I have several concerns about the timing of evaluation for reflux esophagitis. Please see my comments as follows.
<Introduction>
- I strongly recommend rewriting the study rationale focusing on post endoscopic therapy eradication.
<Method>
- Change the word “enrolled” to “collected” and delete the word “consecutive” because your study was retrospective design.
- Delete the sentence “The value has been widely validated in large studies conducted in the Japanese population”.
- When did you evaluate reflux esophagitis after eradication? Also, how long did the evaluation vary between successful eradication and non-successful eradication groups? I have a strong concern the timing of evaluation after eradication. e.g. Successful eradication patients performed early timing of EDG compared with non-successful eradication group. Eighteen patients performed second, third, or fourth-line eradication. When did these patients perform EGD evaluation?
- I disagree the definition of non-successful eradication. I strongly recommend excluding patients with H. pylori seronegative (n =50) because these patients have not current infection.
<Results and Tables>
- Delete all patients data in Table 3 and 4.
- Delete the sentences lines 144-147.
- Please confirm the patient number in newly developed reflux esophagitis in non-successful eradication group.
- I strongly recommend including confounders such as smoking, alcohol use, and aspirin of NSAID use.
Author Response
Response to Queries from Reviewer #1
Comment 1
The following are my comments and recommendations for Drs. Katsurahara and Imoto, on their paper on 'Increased reflux esophagitis after Helicobacter pylori eradication therapy in cases undergoing endoscopic submucosal dissection for early gastric cancer'.
These results were interesting, but I have several concerns about the timing of evaluation for reflux esophagitis. Please see my comments as follows.
Response
We appreciate very much the favorable comments of the Reviewer about our current manuscript.
Comment 2
<Introduction>
I strongly recommend rewriting the study rationale focusing on post endoscopic therapy eradication.
Response
As suggested we have focused the rationale of our present study on post endoscopic therapy eradication.
Therefore, we have made additional statements in the “introduction section”.
Please see page 2, lines 69 to 74 in the revised manuscript.
The additional statement is also described below for your convenience:
“Na et al. reported no increase in the incidence of reflux esophagitis symptoms after H. pylori eradication therapy in patients that underwent endoscopic mucosal resection or endoscopic submucosal dissection for gastric neoplasms [13]. However, no endoscopic study was performed to confirm the presence or absence of reflux esophagitis after eradication therapy, and there is no study performed in a homogenous group of patients with early gastric cancer after endoscopic submucosal dissection.”
Comment 3
<Method>
Change the word “enrolled” to “collected” and delete the word “consecutive” because your study was retrospective design.
Response
We have changed the expression in the text and deleted the word “consecutive”.
Please see page 2, lines 83 to 86 in the revised manuscript.
The section is also described below for your convenience:
“This study comprised 429 patients with gastric cancer admitted to the Department of Gastroenterology and Hepatology, Mie University Hospital, from January 2006 through December 2006. We included 160 patients (males 122, females 38, mean age 69.7 years, range 37-89 years) that fulfilled the study's entry criteria.”
Comment 4
Delete the sentence “The value has been widely validated in large studies conducted in the Japanese population”.
Response
We deleted the sentence.
Please see page 4, lines 148 to 150 in the revised manuscript.
Comment 5
When did you evaluate reflux esophagitis after eradication? Also, how long did the evaluation vary between successful eradication and non-successful eradication groups? I have a strong concern the timing of evaluation after eradication. e.g. Successful eradication patients performed early timing of EDG compared with non-successful eradication group.
Response
As suggested we have clarified when we evaluated reflux esophagitis after eradication and the variation in the time of evaluation between the groups.
Please see pages 3 and 4, lines 123 to 127 in the revised manuscript.
The section is also described below for your convenience:
“The presence and severity of reflux esophagitis were evaluated by esophagogastroduodenoscopy approximately 6 months after eradication therapy in both successful (5.7 ± 2.4 months) and failed (5.6 ± 2.4 months) eradication groups. Another esophagogastroduodenoscopy was performed one year after eradicating H. pylori infection.”
Comment 6
Eighteen patients performed second, third, or fourth-line eradication. When did these patients perform EGD evaluation?
Response
We have also responded to this question of the reviewer.
Please see page 4, lines 139 to 143 in the revised manuscript.
The section is also described below for your convenience:
“A third-line and a fourth-line therapy were also indicated in persistently unresponsive cases. Patients underwent esophagogastroduodenoscopy to evaluate the presence of reflux esophagitis approximately 5 months after the second-line therapy, 10 months after the third-line therapy, and 6 months after the fourth-line eradication therapy.”
Comment 7
I disagree the definition of non-successful eradication. I strongly recommend excluding patients with H. pylori seronegative (n =50) because these patients have not current infection.
Response
Based on the opinion of the reviewer we have clearly separated our original patients into 1) a successful eradication group (n=96), 2) a failed eradication group (n=5), and 3) an untreated control group (n=59). This untreated control group included patients that refused the treatment (n=9) and untreated H. pylori seronegative patients (n=50).
We included the untreated control group to increase the statistical power of our present study, in particular for the univariate and multivariate analysis. We hope that the reviewer agrees with us on this matter.
Based on these changes we have modified Figure 1, and Table 2.
Please also see page 4, lines 164 to 168, and page 6, lines 230 to 233.
Comment 8
<Results and Tables>
Delete all patients data in Table 3 and 4.
Delete the sentences lines 144-147.
Response
As recommended by the reviewer, we have deleted all patients in Tables 3 and 4 and deleted the sentence in lines 144-147 of the original manuscript.
Comment 9
Please confirm the patient number in newly developed reflux esophagitis in non-successful eradication group.
Response
We have clarified the newly developed reflux esophagitis in the group mentioned by the reviewer.
Please see page 6, lines 230 to 233 in the revised manuscript.
The section is also described below for your convenience:
“None of the patients with failed eradication therapy (n=5) developed reflux esophagitis. Four (6.7%) patients (4/59) of the untreated control group had (pre-existing) reflux esophagitis of grade A during the first esophagogastroduodenoscopy, and two (3.3%) patients (2/59) developed reflux esophagitis (grade A or grade B) during the follow-up study.”
Comment 10
I strongly recommend including confounders such as smoking, alcohol use, and aspirin of NSAID use.
Response
As suggested by the reviewer, we have included the confounders pointed out by the reviewer. Please see page 4, lines 168 to 171, Table 2 and Table 3 in the revised manuscript.

Reviewer 2 Report
Dear Authors, congratulations on such an interesting paper. These are several suggestions that I have after reviewing your manuscript:
- Line 68 - 'Gastric' can be written in a small letter. Besides I suppose that there is an additional space before '[12]'. Please remove it.
- Were there any strict inclusion and exclusion criteria regarding patients included in this study? It would be beneficial to add this information clearly in section 2.2.
- Line 285-286 - you have mentioned about the statistics strictly in Japan... Could you kindly provide more general information that could be more representative in terms of general population? And support it by proper references?
- Please check the manuscript once again in terms of English since I have detected several typos or grammatical errors that should be corrected.
Good luck with further research!
Author Response
Response to Queries from Reviewer #2
Comment 1
Dear Authors, congratulations on such an interesting paper. These are several suggestions that I have after reviewing your manuscript:
Response
We appreciate very much the positive and constructive comments of the Reviewer that have improved significantly the quality of the manuscript.
Comment 2
Line 68 - 'Gastric' can be written in a small letter. Besides I suppose that there is an additional space before '[12]'. Please remove it.
Response
We have corrected the errors pointed out by the reviewer.
Please see page 2, lines 66 to 67 in the revised manuscript.
Comment 3
Were there any strict inclusion and exclusion criteria regarding patients included in this study? It would be beneficial to add this information clearly in section 2.2.
Response
As suggested we have clarified the inclusion and the exclusion criteria in section 2.2 of the revised manuscript.
Please see page 3, lines 112 to 117 in the revised manuscript.
The section is also described below for your convenience:
“The exclusion criteria of the study were as follows: current medication with proton pump inhibitors or H2 receptor antagonists during the follow-up study (n=122), lack of follow-up endoscopy (n=74), gastric surgery after endoscopic submucosal dissection (n=46), previous gastric surgery (n=11), or eradication therapy (n=24) (Figure 1). Endoscopic submucosal dissection in early gastric cancer and follow-up by esophagogastroduodenoscopy were the inclusion criteria of the study.”
Comment 4
Line 285-286 - you have mentioned about the statistics strictly in Japan. Could you kindly provide more general information that could be more representative in terms of general population? And support it by proper references?
Response
We have changed the statement in the section, mentioned by the reviewer and added a new reference.
Please see page 7, lines 264 to 266 in the revised manuscript.
The section is also described below for your convenience:
“Although the infection rate by H. pylori has declined in the young population of Japan in recent years, the number of people with increased gastric acid secretion has incremented [16, 17].”
Comment 5
Please check the manuscript once again in terms of English since I have detected several typos or grammatical errors that should be corrected.
Response
We have improved the grammatical errors.